# Application of Whole-Exome Sequencing (WES) for Prenatal Determination of Causes of Fetal Abnormalities

**DOI:** 10.3390/genes16050547

**Published:** 2025-04-30

**Authors:** Margarita Guseva, Natalya Shirokova, Olga Kapitonova, Valentina Gnetetskaya, Konstantin Blagodatskikh, Julia Tarasova, Vladimir Kaimonov, Alina Korbut, Elizaveta Musatova

**Affiliations:** 1The Center of Genetics and Reproductive Medicine “Genetico” JSC, 119571 Moscow, Russia; natalias@genetico.ru (N.S.); kapitonova@genetico.ru (O.K.); blagodatskih@genetico.ru (K.B.); kaimonov@genetico.ru (V.K.); korbut@genetico.ru (A.K.); musatova@genetico.ru (E.M.); 2Clinic “Mother and Child” Savelovskaya, 127015 Moscow, Russia; 3Federal State Budgetary Educational Institution, Further Professional Education “Russian Medical Academy of Continuous Professional Education”, the Ministry of Healthcare of the Russian Federation, 125993 Moscow, Russia

**Keywords:** fetal abnormality, multiple congenital malformations, DNA, next-generation sequencing, whole-exome sequencing, ultrasound, prenatal, diagnosis

## Abstract

Fetal abnormalities are major issues in prenatal medicine. They affect the predicted pregnancy outcome and entail a risk of future recurrent adverse events in a particular couple. In order to clarify the possible outcomes of a specific pregnancy and subsequent ones, it is of the utmost importance to determine the causes of observed fetal abnormalities. Routine laboratory techniques sometimes fail to identify their cause because they are mainly intended for the detection of chromosomal disorders. Over recent years, single-gene disorders have increasingly been regarded as probable causes of fetal abnormalities, and next-generation sequencing (NGS) technologies have been adopted to detect them. This article provides the findings of applying whole-exome sequencing (WES) for prenatal diagnosis. It is aimed at identifying the causes of various fetus abnormalities with a normal molecular karyotype. The diagnostic value of this technique is shown based on the completeness of clinical patterns, abnormality types, and the ability to simultaneously examine the fetus and the parents. Consequently, WES revealed causative variants in 27.27% of cases, which encourages the consideration of applying this technique as part of a state-of-the-art multiple congenital malformation prenatal diagnosis algorithm.

## 1. Introduction

Fetal abnormalities are a pressing challenge in prenatal medicine. The incidence of fetal abnormalities detected by second-trimester ultrasound is 2.3% [1]. The rate of severe fetal congenital malformations associated with a perinatal mortality risk is 1.4% [2]. Once malformations are prenatally detected, a medical professional will inevitably focus on the pregnancy’s outcome, namely, fetal health. However, it is also essential to assess the risk of future recurrent adverse events in the couple. It is often impossible to clarify a potential outcome without determining the causes of observed fetal abnormalities.

Invasive diagnostic techniques, including standard cytogenetic assays, quantitative fluorescence polymerase chain reaction (qPCR), and chromosomal microarray analysis (CMA), are commonly applied to find out the genetic cause of abnormalities. These approaches, designed for analyzing the chromosomal status of a fetus, make it possible to determine up to 40% of abnormalities’ causes. For instance, autosomal trisomies are detected in 30% of cases when either qPCR or a standard cytogenetic assay is performed. The latter also reveals unbalanced structural chromosomal rearrangements in 5% of cases [3]. According to a literature review conducted in 2013, on average, these chromosomal rearrangements are spotted in 6.5% of fetuses with diverse abnormalities once CMA is carried out [4]. Hence, roughly 60% of fetal abnormalities have an unspecified etiology. Some of them are hallmarks of a single-gene disorder. Higher-resolution techniques, such as NGS and massive parallel sequencing (MPS), are required to identify a probable single-gene malformation’s cause [5].

To date, WES has underpinned a series of studies in prenatal medicine. Different authors have stated that the diagnostic value of WES ranges from 9.1% to over 50% for fetuses with malformations [6,7,8,9,10,11,12]. WES sometimes allows one to establish the causes of fetal genetic disorders in prenatal medicine at an early stage, whether or not the phenotypical clinical patterns are complete. This also applies to cases when prenatal phenotypical clinical patterns differ from postnatal ones [13]. Y. Yaron et al. suggest considering exome sequencing as a first-line test in the cases of severe central nervous system (CNS) malformations. The authors have concurred that its diagnostic value is high enough and it enables for the identification of both single-gene causes and non-extended DNA copy number variations. If a fetus is affected by isolated increased nuchal translucency (NT) thickness or other abnormalities, WES may help provide an accurate diagnosis in 3.2–21% of cases [6,9]. One study using WES in prenatal fetus diagnosis with isolated increased nuchal translucency (NT) thickness demonstrated the ability to find out the fetal condition’s cause in 5.5% of cases [14]. A large-scale cohort prospective study published in 2019 analyzed 234 trio-based WES substudies to clarify the causes of fetal abnormalities. The WES diagnostic value was 10% on average and 19% in fetuses with several abnormalities [9]. Data from recent years show that the WES diagnostic value varies from between organ systems. It ranges from 53% in fetuses with skeletal dysplasias to 72% in those with isolated renal hyperechogenicity [15]. A growing number of related publications have raised the issue of possibly revising the approaches to prenatal fetal abnormality diagnosis.

Our study presents the experience of applying WES in fetuses with different abnormalities detected by ultrasound imaging.

## 2. Materials and Methods

We evaluated the biomaterial of 44 fetuses, namely, 44 samples obtained from ongoing pregnancies in the course of invasive prenatal diagnosis. These included 19 samples of chorionic villi, 23 samples of amniotic fluid, and 2 samples of umbilical cord blood. An ultrasound revealed various malformations and/or markers of chromosomal abnormalities in a total of 44 fetuses. The malformations compromised diverse organs and systems. Multiple malformations, skeletal dysplasia, heart defects, limb deformities, and facial abnormalities affected 20, 8, 4, 3, and 2 fetuses, respectively. Brain malformations were detected in 1 fetus. Gastrointestinal malformations were spotted in 1 fetus as well. Moreover, isolated increased nuchal translucency (NT) thickness was noticed in 3 fetuses. Non-immune hydrops affected 1 fetus. NT thickness was also slightly increased in 1 fetus. Once subjected to biochemical screening, one more abnormality (elevated β-hCG levels) was detected.

Trio-based WES involving a fetus and both parents was conducted in 23 out of 44 cases. One fetus and the mother were examined via duo-based WES. The data of the remaining 20 samples were scrutinized in isolation via solo-based WES. The chromosomal status of the fetus was analyzed in 100% of cases, which is considered a first-line test, prior to application of WES. A standard cytogenetic assay, CMA, and both types of analysis were carried out in 3, 28, and 13 cases, respectively. The aforementioned tests did not reveal any chromosomal disorders.

DNA was extracted from peripheral blood by using QIAamp DNA Blood Mini kit (QIAGEN, Hilden, Germany) and from fetal material by using GeneJET Whole Blood Genomic DNA Purification Mini Kit (Thermo Fisher Scientific, Waltham, MA, USA). DNA concentration was measured by means of Qubit dsDNA HS Assay kit and Qubit 2.0 fluorometer developed by Thermo Fisher Scientific. ME220 focused ultrasonicator (Covaris, Woburn, MA, USA) was used to shear DNA into fragments with an average length of 150 bp. SureSelect Human All Exon V7 probe set (Agilent, Santa Clara, CA, USA) assisted in preparing WES libraries in accordance with manufacturer’s manuals. NovaSeq 6000 (Illumina, San Diego, CA, USA) was applied for library sequencing.

Relevant data were processed by an automated algorithm which comprises evaluation of sequencing quality parameters (FASTQC module [16]); removal of adapters and low-quality sequences (SEQPURGE module [17]); alignment of reads to human genome version hg19 (BWA MEM module [18]); filtration of optical and PCR duplicates (SAMBLASTER module [19]); local realignment (ABRA2 module [20]); detection and quality-based filtration of variants (FREEBAYES package [21]); and annotation of variants using clinical information databases (ENSEMBL-VEP module [22]). The following parameters dictate filtration of the detected genetic variants for each sample:(1)The rate does not exceed 2% in the control population sample (gnomad v.2.1.1) [23].(2)A corresponding gene is associated with a specific phenotype in OMIM compendium [24].(3)The variant is located in a coding or splicing region of the gene as per VEP annotation [25].(4)Filtration rules out the variants annotated as benign or likely benign in ClinVar archive [26].(5)The synonymous variants which had not been mentioned in any databases or other literary references before and the variants located in the non-coding regions of the genome were reviewed if the probability of their impact on splicing is high, as evidenced by in silico algorithm data.(6)The variants annotated as pathogenic and likely pathogenic in ClinVar archive were considered without filtration.

The variants were interpreted in compliance with Russian and international clinical guidelines on NGS data interpretation [27,28,29].

## 3. Results

A molecular genetic cause of fetal abnormalities was established in this study when pathogenic or likely pathogenic variants were found in genes associated with a dominant disorder or when said compound heterozygous variants were detected in genes associated with an autosomal recessive disorder, given that the observed phenotype was consistent with the expected one. A possible cause was established in this study when a compound heterozygous pathogenic or likely pathogenic variant was identified on the one allele and a variant of unknown significance (VUS) was found on the other one in cases of a recessive disorder; compound heterozygous VUSs were detected in cases of a recessive disorder; or a VUS was identified in cases of a dominant disorder. The observed phenotype had to be in line with the expected one in all of the above-mentioned scenarios. A cause of the observed fetal phenotype was unspecified in all of the remaining cases.

The fetal DNA WES revealed the causes of abnormalities in 8 out of 44 cases (18.2%, 95% CI 8.61–27.75%). A cause was presumably determined in four other cases (9.09%, 95% CI 1.96–16.21%), as presented in Table 1. Therefore, the detection rate of causative genetic variants amounted to 27.27% (95% CI 16.22–38.31%). The detection rate of causative variants in the solo- and trio-based WES cases reached 30.0% and 26,1%, respectively (Figure 1), although the latter value was expected to be higher. Newly disclosed meta-analysis data show a diagnostic value of 30% for solo-based exome sequencing and of 35% for trio-based sequencing [30]. We also concurred that the detectability of causative variants per se in trio-based cases is concordant with the findings of the study performed by Janicki E. et al., in Belgium [12].

The diagnostic value of WES correlates with the number of detected fetal abnormalities [31].

The detection rate of causative variants depended on the completeness of the phenotypical clinical patterns of the fetus. It varied in our study. The diagnostic value was 35.0% for fetuses with multiple malformations and 20.8% for those with defects of specific organs and systems (Figure 2). These findings are consistent with those of the recent study, where the causes of multiple fetal malformations (29.0%) were more identifiable than the causes of disorders affecting a particular organ or system (16.7%) [10].

According to the literature data, the NGS technique makes it possible to determine the cause in 8% of fetuses with isolated increased NT thickness [32] and in 29% of fetuses with non-immune hydrops [33]. Only three fetuses were affected by the former and one fetus had the latter in our sample. Nonetheless, since the sample had a very small number of said cases, our findings cannot be reconciled with the published data.

The application of WES did not allow a molecular cause of the observed prenatal phenotype to be established in 72.73% of the fetuses with a normal chromosome set (72.73%, 95% CI 58.03–83.77%). 

## 4. Discussion

If standard karyotyping and/or CMA results are normal, WES may contribute to determining the cause of abnormalities in a number of cases. The diagnostic value of this technique constituted 27.27% for fetuses with abnormalities in our study. It is crucial to clarify whether a genetic laboratory requires complete phenotypical clinical patterns or not.

We evaluated the detection rate of causative variants in the solo- and trio-based WES cases. The diagnostic value of the former was found to slightly exceed that of the latter. The difference in the number of detected causative variants may be accounted for by the number of fetuses with multiple abnormalities in our study. There was a total of 55% said fetuses in the solo-based cases, which is superior to those in the trio-based cases (35%), although the expected ratio is the opposite. A smaller percentage of detected causative variants in the trio-based cases compared to that in the solo-based ones may be attributed to the heterogeneous structure and small size of both samples.

The advantages of trio-based WES, which is aimed at determining the cause of a supposed hereditary disease, are the identification of de novo status, objectivization of the inheritance of a particular genetic variant, and determination of cis-/transposition of variants in the genes associated with a recessive disorder. These capabilities have long been embraced in clinical practice. They are especially valued when it is vital to receive the results as fast as possible, e.g., while examining neonates in critical condition [34]. Prenatal medicine implies quick responses to any situation, which helps promptly establish a diagnosis and realize whether it is expedient to prolong the pregnancy or not. Once WES is performed and ultrasound markers are detected, it takes on average 1–2 months to obtain the results.

The completeness of prenatal phenotypical clinical patterns of fetuses is pivotal. Its degree directly influenced the detection rate of causative variants in our study, which did vary. The diagnostic value for fetuses with multiple malformations turned out to be higher than that for those with defects of individual organs and systems. Ultrasound and MRI imaging reveals more complete phenotypical clinical patterns of a fetus, which increases the odds of identifying a genetic cause of malformations.

It is not possible to draw conclusions regarding the frequency of detection of causal variants due to the limited number of fetuses with increased NT and non-immune hydrops in our sample.

## 5. Conclusions

A number of factors, such as a specialist’s qualifications, medical devices, gestational age, fetal position, and specific features of the mother’s habitus, have an impact on the detectability of fetal abnormalities by ultrasound. The most complete phenotypical clinical patterns are often identifiable posthumously. Trio-based WES has a higher diagnostic value than solo- or duo-based WES, engaging a fetus exclusively or a fetus and a single parent. It is more preferable because it allows for the swift identification of de novo variants and cis-/transposition of recessive variants in parents.

The routine application of prenatal WES may be limited by some technical barriers, e.g., the amount and quality of DNA, maternal cell contamination, missing analysis of intronic regions and epigenetic disorders, and different read depths of exonic regions, which entails the risk of false negatives. Ultrasound imaging is likely to poorly reflect the phenotypical clinical patterns, which reduces its diagnostic value. The use of WES may also be impeded by unavailable and unaffordable NGS technologies and a necessity to train specialists in prenatal medicine.

Furthermore, some ethical concerns arise. Exome sequencing may reveal secondary and incidental findings, distinct from the underlying cause of the disease. This has an effect on the future life of the child and other family members. False paternity or consanguineous marriages are occasionally detected. The ownership of test results, including the access of other relatives thereto, is also disputable.

Nevertheless, prenatal WES may come in handy to predict further fetal development, determine the tactics of managing the current pregnancy, assess genetic risks for subsequent pregnancies, and consider ways to prevent them. 

## Figures and Tables

**Figure 1 genes-16-00547-f001:**
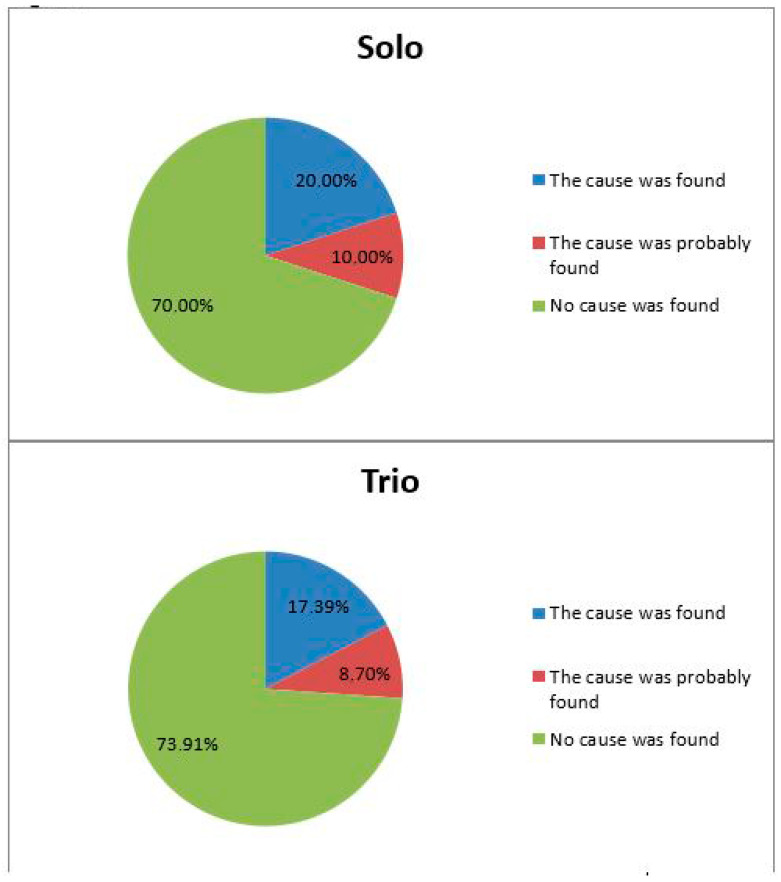
Frequency of detection of causative variants.

**Figure 2 genes-16-00547-f002:**
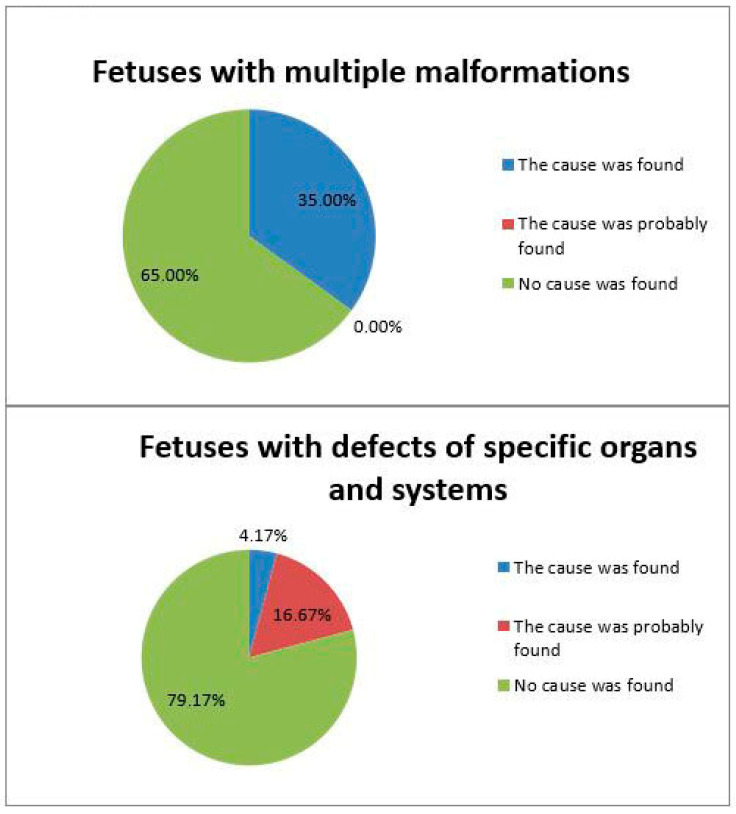
Frequency of detection of causative variants depending on the types of malformations by the number of organs and systems involved.

**Table 1 genes-16-00547-t001:** Variants identified in prenatal cases as causative and possibly causative.

No.	Phenotype	Gestational Age *, Weeks	Gene	Variant 1 (ClinVar)	Variant 2 (ClinVar)	Inheritance **	Solo- or Trio-Based WES
Cases with found cause							
1	Thoracic hypoplasia, abnormal rib shape, short tubular bones	15–16	*FGFR3*	NM_000142.4:c.1948A>G (p.Lys650Glu) (Pathogenic)	-	de novo (AD)	trio-based
2	Heterotaxy syndrome: right-sided heart and stomach	16–17	*MMP21*	NM_147191.1:c.371del (p.Pro124HisfsTer53) (Uncertain significance)	NM_147191.1:c.91C>T (p.Arg31Trp) (Uncertain significance)	genetic variant transposition (AR)	trio-based
3	Increased nuchal translucency, generalized edema of head and trunk, hydrothorax, omphalocele	12–13	*PTPN11*	NM_002834.5:c.226G>C (p.Glu76Gln) (Pathogenic)	-	de novo (AD)	trio-based
4	Cerebellar hypoplasia, ventriculomegaly, brachycephaly	27	*CHD7*	NM_017780.4:c.1611_1612del (Uncertain significance)	-	de novo (AD)	trio-based
5	Lissencephaly, ventriculomegaly, bilateral clubfoot	27–28	*PEX1*	NM_000466.3:c.2489dup (p.Asn830fs) (Likely pathogenic)	NM_000466.3:c.1803+1G>T (Pathogenic)	genetic variant transposition (AR)	solo-based
6	Tuberous sclerosis	30–31	*TSC2*	NM_000548.5:c.4416dup (p.Lys1473fs) (Pathogenic)	-	de novo (AD)	solo-based
7	Cleft lip and palate	28–29	*FGF8*	NM_033163.4:c.33-2A>G (Uncertain significance)	-	n/a (AD)	solo-based
8	Skull shape and left ankle joint features, bilateral femoral campomelia	18–19	*COL1A2*	NM_000089.4:c.2405G>A (p.Gly802Asp) (Pathogenic)	-	de novo (AD)	solo-based
Cases with a possible cause found							
9	Polydactyly postaxial left foot	18	*PTEN*	seq[GRCh37]del(10)(q23.31) chr10:g.89687197_89702520del (Uncertain significance)	seq[GRCh37]del(10)(q23.31) chr10:g.89721087_89728496del (Uncertain significance)	n/a (AD)	trio-based
10	Femoral shortening and curvature	21–22	*FBN1*	NM_000138.5:c.6291_6292delGCinsTT (p.Glu2097_Leu2098delinsAspPhe) (Uncertain significance)	-	n/a (AD)	solo-based
11	Right-sided cleft lip and palate	16–17	*EVC*	NM_153717.3:c.2278C>T (p.Arg760Trp) (Uncertain significance)	NM_153717.3:c.1019G>A (p.Arg340Gln) (Uncertain significance)	genetic variant transposition (AR)	trio-based
12	Equinovarus deformity of the feet	18–19	*NEB*	NM_001164508.2:c.21796C>T (p.Pro7266Ser) (Uncertain significance)	NM_001164508.2:c.19626+63A>T (Uncertain significance)	genetic variant transposition (AR)	solo-based

* Gestational age at which fetal material was taken, ** inheritance was determined according to trio analysis and/or Sanger confirmatory sequencing results. n/a—not available.

## Data Availability

The original contributions presented in this study are included in the article. Further inquiries can be directed to the corresponding author(s).

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
