# Peer review of "Application of Whole-Exome Sequencing (WES) for Prenatal Determination of Causes of Fetal Abnormalities"

_genes, 2025, doi:10.3390/genes16050547_

Round 1
Reviewer 1 Report
Comments and Suggestions for Authors
Guseva et al describe use of exome sequencing for prenatal determination of fetal abnormalities in 44 fetuses with ultrasound evidence of fetal malformations in Moscow, Russia.
The use of this technology for the purpose of prenatal analysis is not novel; however this Russian group summarize their diagnostic yield and how this compares to other published reports. The overall detection rate of 27.27% is consistent with that reported in other studies. This report adds to the medical literature on use of NGS analysis for prenatal testing.
Comments for improvement:
- In the Table, please include the variant interpretation for all of the reported variants. The variant interpretation is included in parentheses for some of the variants, but others state (n/a). It is not clear in the manuscript why variant interpretation is (n/a) for some of the reported variants.
- It has become standard in the field to use the term “exome sequencing (ES)” rather than “whole exome sequencing (WES).” I do not know whether the journal Genes has any recommendations for this terminology.
- Please review the recent literature and include recent seminal publications regarding the use of exome sequencing for prenatal diagnosis, including: 1. Brand et al NEJM 2023 Correspondence on use of trio exome sequencing of cfDNA as noninvasive fetal screening and 2. Lowther et al ASHG 2023 regarding the diagnostic yield of whole genome sequencing.
- In the discussion, lines 175-177: please reword this sentence. It is unclear what point the authors are trying to make.
- Please reword the last sentence of the Discussion (lines 202-204). The authors state “it is impossible to draw conclusions regarding detectability of causative variants due to a limited number of fetuses…” Are the authors referring to the detection rate specifically in fetuses with increased NT and hydrops or in all fetuses?
Author Response
Comments for improvement:
- In the table we have added a variant interpretation of all the presented options.
- Thank you for the clarification. For now we have left the term "whole exome sequencing (WES)". If necessary, we will make corrections
- We will soon get acquainted with the literature you recommend. It should be very interesting.
- We have made a clarification to the text
- We have made a clarification to the text
Reviewer 2 Report
Comments and Suggestions for Authors
The authors present a methodologically sound and good manuscript, however there are some points that should be corrected before publishing.
- The authors decribe in the introduction (line 47) a literature review conducted in 2013. That does not soudn very up to date, so either they conduct a fresh brief review themselves with two to rather four fresh references or find some more recent literature regarding that topic.
- The authors use QF-PCR instead of qPCR as a standard abbreviation for real-time PCR. Please use qPCR throughout the manuscript, which is international publishing standard.
- The authors also use multiple introductions of abbreviations, altough they have been already introduced like nuchal transluceny (NT) in lines 62 ans 84, respectively. One introduction should be enough.
- In the methods section, the authors point out non-coding regions of the genome (line 119). As whole exome is analysed, I think the authors mean intronic sequences or sequences of the promoter region mostly, that are still covered by the WES assay used, however this should be more clearly stated.
- The conclusions are a bit blurry, thus how the findings could lead to novel insights could be more elaborated in the section, like the use os cell-lines and knock-out models for instance.
The manuscript should be proof-read and corrected by a native speaker, as all parts of the manuscript could profit from a bit of improvement.
Author Response
- Thank you for your comments. We have added links to more recent literature on this topic.
- We replaced QF-PCR with qPCR
- We have removed the repetition of the full name
- We have clarified the term non-coding regions of the genome